# Effects of Fruit Load on Sugar/Acid Quality and Puffiness of Delayed-Harvest Citrus

**Fangjie Xu** [1], **Haishan An** [1], **Jiaying Zhang** [1], **Zhihong Xu** [2,*] and **Fei Jiang** [3]

1   Shanghai Key Lab of Protected Horticultural Technology/Forestry and Pomology Research Institute, Shanghai Academy of Agricultural Sciences, No. 1000 Jinqi Road, Fengxian District, Shanghai 201403, China; funjx@163.com (F.X.); anhaishan@saas.sh.cn (H.A.); jiayingzsaas@163.com (J.Z.)

2   College of Advanced Agricultural Sciences, Zhejiang A&F University, No. 666 Wushu Street, Lin'an District, Hangzhou 311300, China

3   Shanghai Research Institute of Citrus, No. 1888 Fenghuang Road, Changxing Island, Shanghai 201913, China; jflqw@126.com

*   Correspondence: zhhxu@zafu.edu.cn; Tel.: +86-571-637-412-70

**Abstract:** Delayed harvesting technology is believed to improve the citrus fruit flavor, but improper tree fruit load under delayed harvest might cause puffiness and reduce fruit quality. In order to find out an optimum tree fruit load level to obtain better flavor quality as well as reduce puffiness in delayed-harvest citrus under protected cultivation, experiments were conducted in the present study between 2019 and 2020 to determine the effect of different fruit loads and fruit-bearing per single branch on the soluble sugars and organic acids metabolism in the peel and flesh, the anatomical structure of the matured fruit peel, and fruit texture-related indexes. The results suggested significant negative correlations between leaf N level and flesh sucrose and glucose contents, and between branch P level and flesh citric acid contents; no significant correlation between NPK levels and flesh texture; relatively lower leaf N and branch P under relatively higher load can increase flesh sucrose and glucose accumulation and slow down citric acid degradation to the greater extent, thus optimizing the sugar/acid ratio of fruits during delayed harvest. The lignification of parenchyma cells closely around peel secretory cavities due to ascorbic acid deficiency might be the primary cause for puffiness under low-load treatments.

**Keywords:** satsuma mandarin; tree fruit load; sugar/acid quality; puffiness; fruit texture analysis; delayed harvest



## 1. Introduction

China is the world's largest producer of Satsuma mandarin (*Citrus unshiu* Marc.), accounting for more than 40% of the world's total production [1]. Shanghai is located on the northern edge of a suitable citrus-planting area in China; due to insufficient annual accumulated temperature, Shanghai's local citrus crop exhibits insufficient sugar accumulation, high acidity and delayed maturity, which severely limits its quality. To solve this problem, 'Oita wase', a very early-maturing variety of satsuma mandarin, together with delayed-harvest technologies, has been introduced into Shanghai in recent years and has attracted increasing attention [2].

Soluble sugar and organic acid levels in the flesh are the two major fruit quality determinants. The major soluble sugar and organic acid associated with flesh sweetness and acidity is sucrose and citric acid, respectively [3]. The fruit quality formation period of citrus could be divided into three phases: the fruit rapid-swelling stage (2.5–3.5 months after flower bloom, phase I), peel color-changing stage (3.5–4.5 months after flower bloom, phase II) and fruit ripening stage (4.5–5 months after flower bloom, phase III). Phase III is critical to the rapid accumulation of soluble sugar/organic acid, and formation of the unique flavors of citrus [4]. During phase III, the rate of fruit volume increase is greatly

reduced. Externally, the major change is color break, and internally, sugar and acid levels reach the desired levels for harvesting and consumption [3].

In most cases, delayed harvest could prolong the duration of phase III of fruit development, further increase the total soluble solid (TSS) content, and optimize the TSS/TA ratio of fruits. However, under the condition of delayed cultivation, inappropriate fruit load level (numbers of fruit set in the whole tree) may aggravate the fruit puffing and reduce the fruits' sugar/acid quality [5,6]. Therefore, it is necessary to study the soluble sugar and organic acid metabolism of the flesh and peel separately under different levels of tree fruit load, and to reveal their possible relationships with puffiness (different degrees of peel-flesh separation caused by over ripening of citrus fruits) to ensure high quality fruits are maintained even in a delayed harvest mode.

In addition to content and components of TSS and TA, the fruit mastication trait also plays a decisive role in the formation of internal quality of fruits, and is related mainly with dietary fiber, especially pectin [7]. Cell walls are composed of cellulose, hemicellulose, pectin, and sometimes lignin, and pectin is one of the important compounds that harden the cell wall. When it is degraded by enzymes during fruit maturation, tissue softens gradually. Cellulose and hemicellulose are also degraded by cellulase during this period, and finally fruit sweetness and mastication traits were improved. The fruit mastication characteristics can be quantified using a fruit texture analyzer [8].

Most traditional researchers believed that puffiness was due to albedo cell rupture in peel secretory cavities [9], while the rupture of albedo cells may be caused by water imbalance (drought, excessive precipitation, high humidity) [4,10], intracellular imbalance of redox status [11,12], metabolic disorders of endogenous hormones [13,14] or calcium (Ca) deficiency [15]. However, to date, the soluble sugar and organic acid metabolism in peel and its relationship with fruit texture formation still needs to be elucidated. For this purpose, the effects of different fruit loads of tree bodies and single branches on soluble sugar and organic acid distributions in flesh and peel during the fruit ripening period were evaluated in the present study.

## 2. Materials and Methods

### 2.1. Plant Material and Experimental Design

Four-year-old satsuma mandarins (*Citrus unshiu* Marc. cv. 'Oita wase') were used as the test material (using *Citrus trifoliata* (*Poncirus trifoliata* (L.) Raf.) as rootstocks, all tested trees started to bear fruit in 2018). The planting distance was 4 m × 4 m. In order to make the difference of puffiness degree more obvious among different treatments, greatly varied tree fruit load levels were applied, including the control (CK, 120 fruits per tree), 20 fruits per tree (L-20), and 30 fruits per tree (L-30). Furthermore, branches with different fruit-bearing were taken as the samples: CK-0 (branch with no fruit-bearing), CK-1 (one fruit per branch), CK-2 (two fruits per branch), CK-4 (four fruits per branch). The same treatments were also applied under low-load treatments (L-20-0, L-20-1, L-20-2, L-30-0, L-30-1, L-30-2). The experimental design consisted of a randomized block with 12 replications, and each replicate represented one tree.

The experiment was carried out during 2019–2020 in a greenhouse located in Zhuang Hang town, Feng Xian district, Shanghai Academy of Agricultural Sciences in Shanghai, China (116°41′33″ N, 39°91′09″ E, annual average temperature 15.8 °C, annual precipitation 1161.1 mm). Temperature settings in the greenhouse were: 28 °C/17 °C (±5 °C, day/night) in spring, summer and autumn, and 6 °C/3 °C (±2 °C, day/night) in winter, relative humidity about 70 ± 5%. Fertilizer application schedule: 0.5 kg of water-soluble fertilizer (N:P:K = 30:15:15) per tree provided every two weeks from March to May; 0.5 kg of water-soluble fertilizer (N:P:K = 15:40:40) per tree provided every two weeks from June to August; 0.5 kg of water-soluble fertilizer (N:P:K = 15:30:30) per tree provided once a week from September to November; and 10 kg of organic fertilizer per tree given as basal fertilizer in December. During the fruit swelling period (7.10th–8.20th), additional foliar trace element

fertilizers were applied twice. The water supply was delivered according to the actual needs of the trees using auto sprinkling irrigation system.

## 2.2. Analysis of Basic Physical and Chemical Properties of Orchard Topsoil

The orchard soil samples were collected from nine randomly selected points in the greenhouse before and after fruit harvest; 1 kg of soil samples per point were collected from different soil layers (0–20 cm and 20–40 cm, respectively), mixed and divided into three parts, and then air-dried, ground and passed through a 2 mm sieve for the determination of basic physical and chemical properties [16]. The content of total nitrogen (TN) was determined by the Kjeldahl method, total phosphorus (TP) content was determined by the colorimetric method with ammonium molybdate, and the total potassium (TK) content was determined by fame photometry [16]. The soil organic matter (SOM) was determined according to the method of Walkley and Black [17].

## 2.3. Determination of NPK Contents in Branches and Leaves

Mature leaves of the same age and annual spring branches with or without fruit-bearing were selected as test samples in the present study, the content of total nitrogen (TN), total phosphorus (TP) and the total potassium (TK) content in branches and leaves were determined according to the method reported by Wang et al. [18]. The sampling duration lasted from the early stage of fruit ripening to the fully-ripening stage (three time points: mid-September, early-October, early-mid-November).

## 2.4. Fruit TSS Content and Weight Measurement

The total soluble solid (TSS, %) content of fruit was measured by a hand-held refractometer, the changes of average weight of fruit (AWF) were measured by electronic precision balance. Presented results are means of 10 replicates per treatment.

## 2.5. Determination of Soluble Sugar and Organic Acid Contents and Components

Agilent 1100 high-pressure liquid chromatography (HPLC) system (Agilent Technologies, Santa Clara, CA, USA) was used. A CARBOSep CHO-620 capillary column (10 μm × 6 mm × 250 mm) (Transgenomic, Inc., New Haven, CT, USA) and a differential refraction detector were used for soluble sugar (sucrose, fructose, glucose) determination. The column temperature used was 80 °C, and the mobile phase consisted of $ddH_2O$. A ZORBAX Eclipse XDB-C185 capillary column (10 μm × 4.6 mm × 250 mm) (Agilent Technologies, Santa Clara, CA, USA) and an ultraviolet detector were used for organic acid (citric acid (CA), malic acid (MA)) determination. The detector was set at $\lambda$ = 210 nm, and the mobile phase was a 0.02 M $KH_2PO_4$ solution (pH = 2.9) [16]. Ascorbic acid (AsA) content was determined following the method described by Zhou et al. [19].

## 2.6. Determination of Fruit Firmness and Hardness

Fully-matured fruits (harvested in the midNovember) were used for textural analysis; a texture analyzer (TA. XTplus, Stable Micro Systems, Godalming, UK) with P/2 probe (cylindricality, 2 mm in diameter) was used to perform a puncture test. The test speed of the probe was 1 mm/s to a depth of 10 mm; the pretest speed was 5 mm/s, and the posttest speed was 5 mm/s. The minimum trigger force was 5 *g*, the threshold force was 2 *g* and the data acquisition rate was 400. Twelve replicates were conducted for each treatment, and then the results were averaged [8].

## 2.7. Anatomical Observation of Peel Structure

The peel anatomical structure of fully-matured fruits was observed by method of paraffin cut, and the samples embedded in paraffin were sliced (8 μm) with a microtome (Leica RM2265, Leica Biosystems Nussloch GmbH D-69226 Nussloch, Germany), stained with safranine-fast green, and then examined under a microscope (Nikon Eclipse E200MV RS NIKON Corporation, Tokyo, Japan).

*2.8. Statistical Analysis*

Data and statistical analysis, including Student's t-test and ANOVA, principal component analysis (PCA) and correlation analyses were carried out via SPSS 18.0 (SPSS Inc., Chicago, IL, USA). Differences were considered statistically significant at $p < 0.05$, and graphing was performed using Prism 4 (GraphPad, Lo Jolla, CA, USA).

**3. Results**

*3.1. The Physical and Chemical Properties of Orchard Soil before and after Harvest*

As shown in Table 1, the EC values and the contents of available NPK and SOM after harvest were significantly lower than those of before harvest. Furthermore, these values in the 0–20 cm topsoil layer decreased more than those in the 20–40 cm soil layer, which suggested that most of soil mineral nutrients that could be efficiently utilized by trees were predominantly in the 0–20 cm soil layer.

**Table 1.** Before (mid-September) and after harvest (late-November), the basic physical and chemical properties of soil samples of different soil layers (0–20 cm and 20–40 cm) were measured, including pH value, electrical conductivity (EC) value, available nitrogen (N) content, available phosphorus (P) content, available potassium (K) content and soil organic matter (SOM) content.

| Test Time | Soil Layer | Soil pH | EC (μs/cm) | Available N (mg/kg) | Available P (mg/kg) | Available K (mg/kg) | SOM (%) |
|---|---|---|---|---|---|---|---|
| Before harvest (mid-September) | 0–20 cm | 7.11 ± 0.06 [d] | 265.32 ± 106.28 [b] | 92.60 ± 23.14 [a] | 63.28 ± 11.54 [a] | 268.14 ± 26.14 [a] | 1.59 ± 0.48 [a] |
| | 20–40 cm | 7.45 ± 0.04 [c] | 316.47 ± 84.44 [a] | 74.22 ± 11.65 [b] | 47.82 ± 16.37 [b] | 169.64 ± 24.32 [b] | 1.17 ± 0.42 [b,c] |
| After harvest (Late November) | 0–20 cm | 7.63 ± 0.12 [b] | 187.02 ± 67.26 [c] | 72.27 ± 10.85 [b] | 46.82 ± 13.58 [b] | 96.65 ± 18.92 [c] | 1.29 ± 0.32 [b] |
| | 20–40 cm | 7.85 ± 0.25 [a] | 258.53 ± 63.32 [b] | 65.65 ± 22.16 [c] | 35.86 ± 10.23 [c] | 78.32 ± 15.06 [d] | 0.97 ± 0.12 [d] |

Within columns, means with the different letter indicate significant differences ($p < 0.05$) using Duncan's test. Standard errors of means of 3 replicates. The time point of soil collection: Before harvest (mid-September): during the period of rapid accumulation of soluble sugar/organic acid of fruits; After harvest (late-November): during the time period of that after fruit harvest and before the fertilization in winter (late-December). The data is the average value of two consecutive years.

*3.2. Effect of Fruit Load Treatment on NPK Content in Branches and Leaves*

As shown in Figure 1, due to the sufficient fertilization in December 2019, the tree nutrient level, especially P and K content in both branch and leaf in 2020, was much better than that in 2019 in whole. Leaf N content decreased slightly and continuously with fruit ripening; the decrement of leaf N in 2020 was relatively greater than that of 2019. Compared to the low-load treatments, the leaf N content decreased even more under treatment of CK. Furthermore, CK-2 and CK-4 performed a more significant decrease than that of CK-0 and CK-1, which indicated that most of the nitrogen needed in fruit developing was supplied directly by leaves. Higher tree fruit load and higher fruit bearing per single branch consumed more leaf N nutrient with fruit development as compared to those low-load treatments.

Meanwhile, Figure 1 showed that the leaf P and branch NP content changed very little in 2019, while in 2020, leaf P decreased more significantly under the low-load treatments as compared to that of CK. Branch NP showed an opposite trend, that it decreased more under the treatment of CK with fruit development, especially in CK-2 and CK-4. Furthermore, the P requirement during fruit development might come from not only leaves but also branches. The opposite trend of leaf P and branch P might indicate that relatively higher tree fruit loading and higher fruit bearing per single branch can improve the utilization of tree-stored nutrients (in branch). The variation of K content in branches and leaves was quite different between these two years. In 2019, branch K content under the treatment of CK decreased slightly and constantly while that under low-load treatments kept stable (L-20) or even slightly increased (L-30); the largest decrease in branch K content was observed in CK-4 in October. In 2020, branch K decreased significantly and constantly under the treatment of CK, and the decrement was greater during the later stage of fruit ripening (early October to mid-November) than that of the early stage (mid-September to early October). Meanwhile, the decrements were greater in the low-load treatment of L-30 than that of L-20.

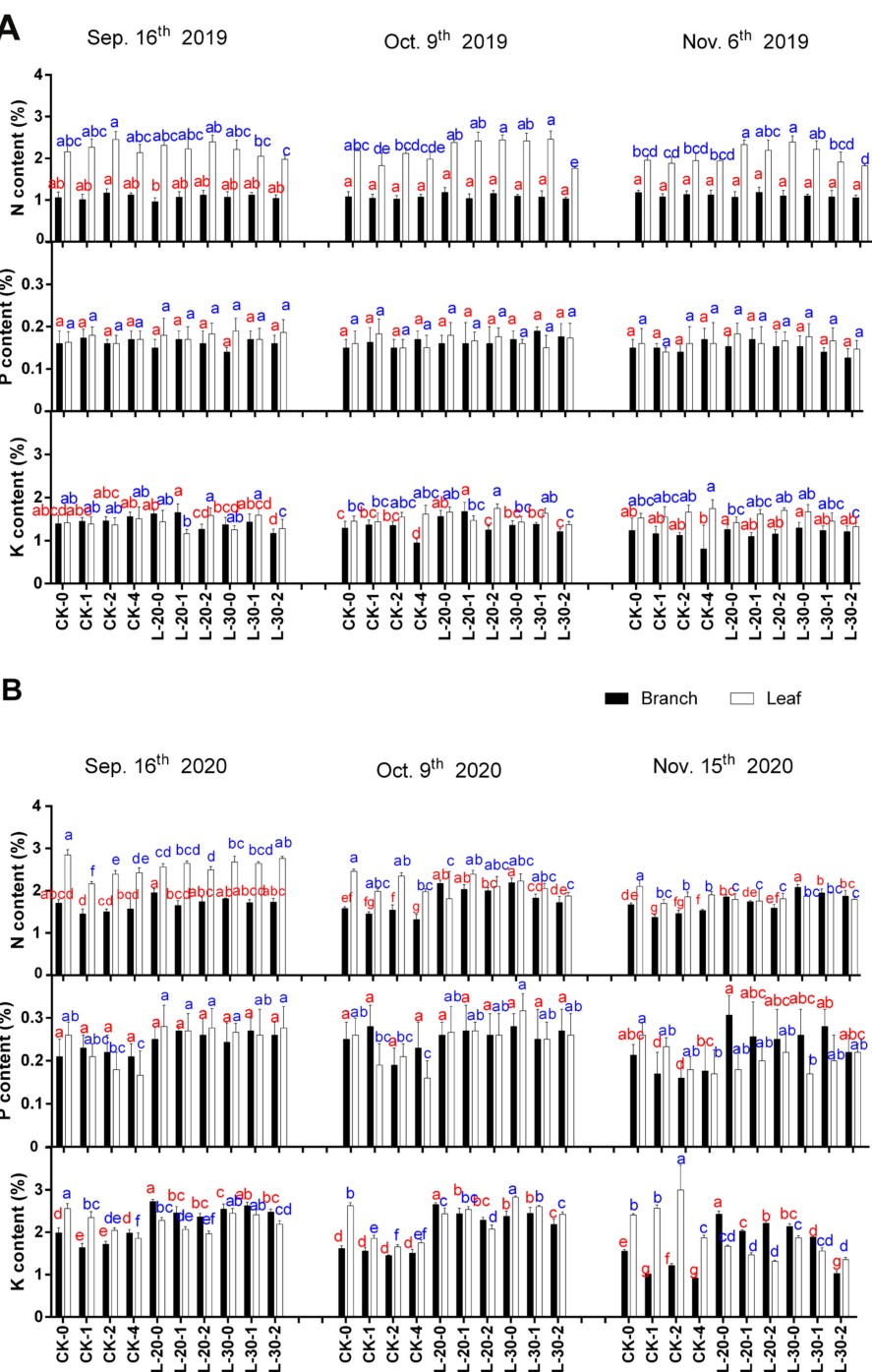

**Figure 1.** Comparison of changes of NPK contents in branches and leaves under different fruit load treatments from the near-mature period (mid-September) to the fully-mature period (mid-November) during 2019 (**A**) to 2020 (**B**). Within columns, means with the different letter indicate significant differences ($p < 0.05$) using Duncan's test; bars on the columns represent the standard error of the mean ($n = 3$).

In 2019, leaf K content remained stable under treatments of CK and low-load treatments, while in 2020 K content, both in branches and leaves, decreased slightly at first and then decreased significantly during the later period of fruit ripening; the branch K content of CK treatment decreased more than low-load treatments. All of these results might indicate that under relatively higher tree fruit load and fruit-bearing level, compared

with N and P, the contribution of K for fruit development may be more obvious, especially in the delayed harvest period.

In general, the decrease of NPK content in branches and leaves with fruit development in 2020 is much larger than that in 2019, especially under the treatment of CK, which indicated that higher tree fruit load level for two consecutive years might cause too much consumption of tree stored nutrients, especially branch NP. This may also be one of the reasons for the inconsistent trend of tree nutrient data changes during fruit development in the two consecutive years (2019–2020).

### 3.3. Effect of Fruit Load Treatment on Fruit Weight and TSS Development

Table 2 showed that the TSS and AWF of CK-1 increased by 38.2% and 41.8% with fruit development, respectively, which was higher than that of other treatments (the increments were 24.4~31.3% and 4.6~33.9%, respectively). These results suggested that within the treatment of CK, treatments with relatively lower fruit bearing per branch (CK-1 or CK-2) showed higher TSS contents and higher AWF as compared to that treatment with higher fruit bearing per branch (CK-4); too low of a tree fruit load was not conducive to the accumulation of TSS during fruit ripening.

### 3.4. Effect of Fruit Load Treatment on Fruit Sugar/Acid Content and Component

As shown in Figure 2, in 2019 the contents of glucose and fructose in peels under both CK and low-load treatments kept stable, while the content of sucrose in peels decreased continuously in CK-4 and low-load treatments with fruit development. In flesh, the treatment of CK showed a slight and constant increase in the content of sucrose, and CK-2 presented the highest increments. In 2020, the contents of sucrose, fructose, and glucose in peels under low-load treatment of L-20 kept stable with fruit development, while low-load treatment of L-30 led to a constant decrease in peel sucrose and glucose.

In flesh, by contrast, in 2019 sucrose, fructose and glucose content under the treatment of CK-1 and CK-4 remained stable with fruit development, CK-2 showed a great increase in sucrose and glucose content, while under the low-load treatments (L-20 and L-30) fructose content increased constantly with fruit development while the contents of sucrose and glucose decreased at first and then increased slightly during the delayed harvest period. The inflection point of all treatments appeared around mid-October.

The variation of contents of three soluble sugars in both peel and flesh in 2020 was quite similar to that in 2019. All these results suggest that the further accumulation of sucrose in flesh promoted by delayed harvesting can be realized under relatively higher fruit loads, while the contents of fructose and glucose changed slightly, which suggested that delayed harvesting could further increase the sucrose content to a greater extent in the treatment of CK.

Figure 3 showed that the contents of citric acid and malic acid in peel and flesh showed different developing trends throughout fruit ripening. In 2019, the contents of peel citric acid and malic acid decreased more under low-load treatments as compared to that of CK, especially during the delay harvesting period (early October to early November). Meanwhile, under the treatment of CK, the content of flesh citric acid kept stable with fruit development, and the content of flesh MA showed an increase first and then decreased slightly. Under the low-load treatments, the flesh citric acid and malic acid showed a similar trend of a continuous decrease. All of these results suggest that low-load treatments decreased the contents of flesh citric acid and malic acid to a greater extent as compared to that of CK, which indicated that under the treatment of CK, fully-ripened fruits maintained relatively higher total acidity than that of low-load treatments. The change trend of organic acids in 2020 was similar to that in 2019; to some extent, different fruit load levels set in the present study did change the content and proportion of different types of organic acids, especially the ratio of citric acid and malic acid contents in the flesh.

**Table 2.** The fruit weight increase and TSS accumulation under different fruit load treatments were determined from near-mature period (mid-September) to fully-mature period (mid-November).

| Sampling Time | Indicators | CK-1 | CK-2 | CK-4 | L-20-1 | L-20-2 | L-30-1 | L-30-2 |
|---|---|---|---|---|---|---|---|---|
| 2019.9.16 | AWF (g) | 72.4 ± 5.47 [e] | 92.87 ± 6.57 [c] | 86.7 ± 5.63 [c,d] | 155.3 ± 11.04 [a,b] | 167.67 ± 15.62 [a] | 148.71 ± 8.65 [b] | 150.46 ± 10.32 [b] |
| | TSS (%) | 7.83 ± 0.25 [b] | 7.96 ± 0.24 [a,b] | 8.53 ± 0.16 [a] | 7.65 ± 0.23 [b,c] | 7.54 ± 0.14 [c] | 7.63 ± 0.02 [b,c] | 7.67 ± 0.06 [b,c] |
| 2019.10.9 | AWF (g) | 92.69 ± 6.58 [d,e] | 91.65 ± 9.52 [d,e] | 98.15 ± 8.69 [d] | 182.92 ± 15.42 [a] | 176.84 ± 10.65 [a,b] | 162.1 ± 12.58 [b,c] | 167.16 ± 6.85 [b,c] |
| | TSS (%) | 9.25 ± 0.25 [a,b] | 9.45 ± 0.21 [a] | 9.06 ± 0.16 [b,c] | 8.54 ± 0.24 [c] | 8.95 ± 0.21 [b,c] | 9.03 ± 0.16 [b,c] | 9.24 ± 0.28 [a,b] |
| 2019.11.6 | AWF (g) | 102.64 ± 8.24 [d,e] | 105.74 ± 6.54 [d,e] | 119.51 ± 8.69 [d] | 192.50 ± 18.62 [a] | 175.35 ± 14.24 [b] | 159.77 ± 10.21 [b,c] | 162.36 ± 5.32 [b,c] |
| | TSS (%) | 10.82 ± 0.24 [a] | 10.45 ± 0.12 [a,b] | 11.02 ± 0.21 [a] | 9.65 ± 0.19 [c] | 9.86 ± 0.17 [b,c] | 10.02 ± 0.21 [b] | 9.54 ± 0.24 [c] |
| 2020.9.16 | AWF (g) | 76.4 ± 4.65 [c] | 87.56 ± 6.53 [c] | 80.34 ± 6.23 [c] | 136.24 ± 10.21 [a,b] | 125.62 ± 9.73 [b] | 151.25 ± 9.25 [a] | 140.35 ± 8.46 [a,b] |
| | TSS (%) | 7.63 ± 0.34 [b] | 8.06 ± 0.52 [a,b] | 8.36 ± 0.43 [a] | 7.35 ± 0.32 [b,c] | 7.78 ± 0.36 [b] | 7.32 ± 0.45 [b,c] | 7.49 ± 0.37 [b,c] |
| 2020.10.9 | AWF (g) | 95.78 ± 6.58 [c] | 92.46 ± 4.86 [c,d] | 100.25 ± 6.79 [c] | 168.12 ± 13.64 [a] | 134.24 ± 9.25 [b] | 157.64 ± 9.58 [a,b] | 143.67 ± 8.25 [b] |
| | TSS (%) | 9.65 ± 0.34 [a] | 9.55 ± 0.35 [a,b] | 9.78 ± 0.34 [a] | 9.01 ± 0.42 [b] | 8.75 ± 0.32 [b] | 8.96 ± 0.46 [b] | 8.65 ± 0.24 [b,c] |
| 2020.11.15 | AWF (g) | 112.63 ± 5.88 [d] | 102.36 ± 7.32 [d,e] | 103.68 ± 10.01 [d,e] | 187.06 ± 11.42 [a] | 155.65 ± 9.34 [b,c] | 168.47 ± 8.21 [b] | 154.86 ± 7.32 [b,c] |
| | TSS (%) | 10.32 ± 0.44 [a,b] | 10.85 ± 0.32 [a] | 11.52 ± 0.46 [a] | 9.45 ± 0.64 [c] | 9.86 ± 0.46 [b,c] | 10.24 ± 0.41 [a,b] | 9.77 ± 0.35 [b,c] |

Within the same time point, means with the different letter indicate significant differences ($p < 0.05$) using Duncan's test. Standard errors of means of 12 replicates.

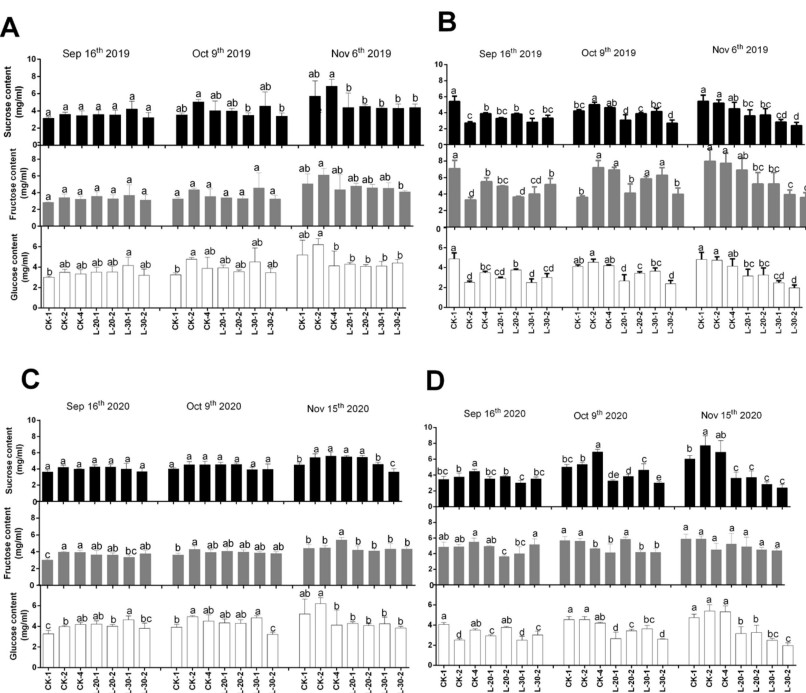

**Figure 2.** Comparison of changes of soluble sugar components and contents in peel (**A**,**C**) and flesh (**B**,**D**) under different fruit load treatments from the near-mature period (mid-September) to the fully-mature period (mid-November). Within columns, means with the different letter indicate significant differences ($p < 0.05$) using Duncan's test, bars on the columns represent the standard error of the mean ($n = 3$).

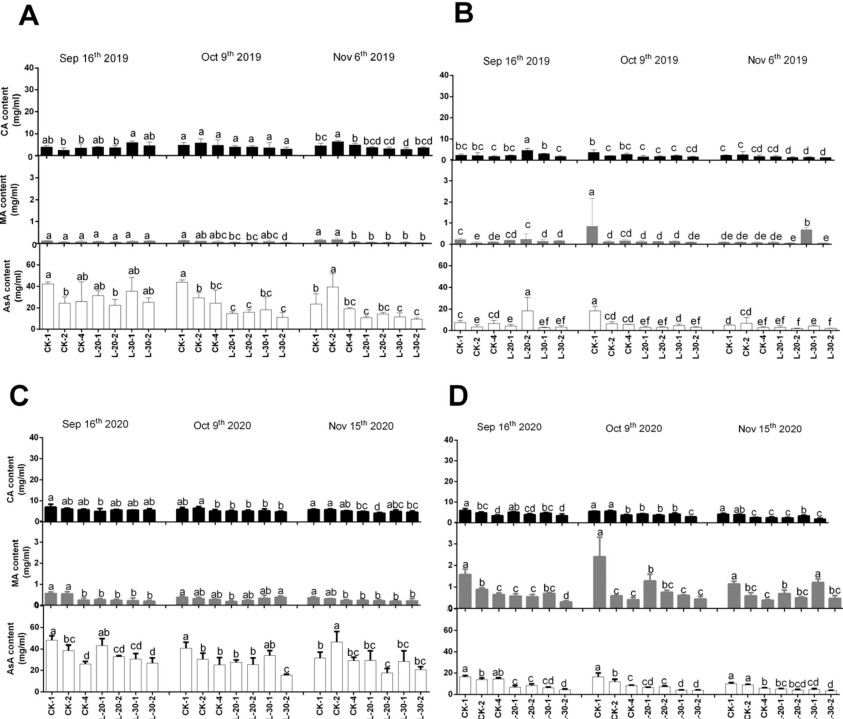

**Figure 3.** Comparison of changes of organic acid components and contents of peel (**A**,**C**) and flesh (**B**,**D**) under different fruit load treatments from the near-mature period (mid-September) to the fully-mature period (mid-November) (CA: citric acid; MA: malic acid; AsA: ascorbic acid). Within columns, means with the different letter indicate significant differences ($p < 0.05$) using Duncan's test; bars on the columns represent the standard error of the mean ($n = 3$).

The ascorbic acid (AsA) content in peel and flesh also performed a similar trend in 2019 and 2020: the peel AsA content under the low-load treatments kept stable (L-20) or decreased slightly (L-30) with fruit development, while under the treatment of CK, the peel AsA decreased in CK-1, and increased in CK-2 and CK-4 with fruit development. Furthermore, during the delayed harvest period (early October to early-mid November), changes of AsA content in both peel and flesh changed more significantly under treatment of CK than that of low-load treatments, it could be concluded that during the delayed harvest period, the further degradation of AsA in the peel and flesh was a relatively independent process. The relatively lower and stable peel and flesh AsA content under lower-load treatments might indicate insufficient antioxidant capacity, which might be one of the reasons for the occurrence of puffiness.

### 3.5. Effect of Fruit Load Treatment on Peel Anatomical Structure

As shown in Figure 4, there were no obvious abnormal changes in the secretory cavity structure of the peel under the CK treatment, while under the low-load treatments, although the structure of secretory cavities were still normal, a small part of the parenchyma cells in the flavedo layer between the two adjacent secretory cavities of the peel were deformed (L-20) or even ruptured (L-30) due to extrusion caused by peel deformation.

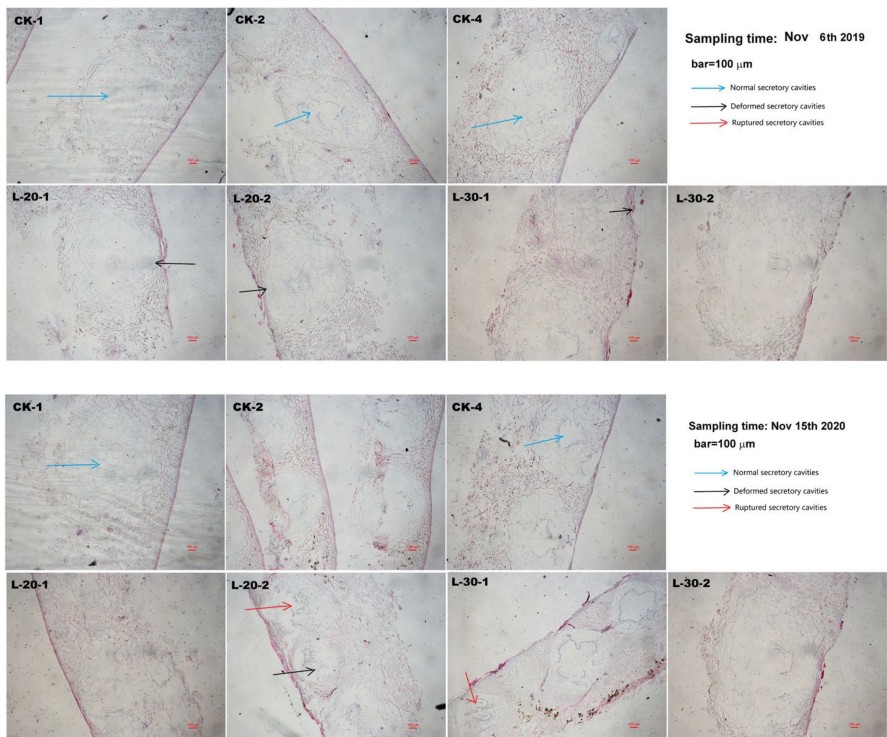

**Figure 4.** Observations of the cross-sectional anatomical structures of peel of fully-matured fruits (Sampling time: 6 November 2019 and 15 November 2020) under different fruit load treatments (bar = 100 μm).

### 3.6. Effect of Fruit Load Treatment on Fruit Texture

Figure 5 showed that the low-load treatments resulted in relatively higher flesh hardness, segment hardness, and peel toughness than those of the treatment of CK, which indicated that the lower-load treatments obviously worsened fruit mastication. In the CK treatment, relatively higher fruit-bearing per branch reduced peel hardness slightly and improved peel toughness significantly, but had no obvious effect on segment hardness or firmness. This suggested that under higher tree fruit load, relatively higher fruit-bearing per branch might lead to thinner, softer, but flexible peels, but its effect on improving fruit mastication was limited.

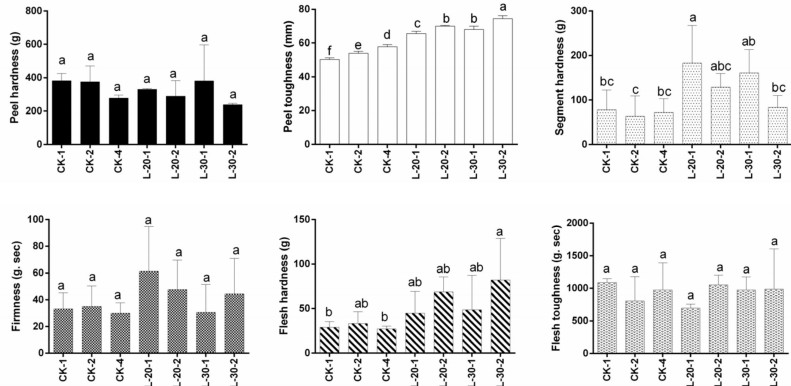

**Figure 5.** Textural analysis of fully-matured fruits (Sampling time: 15 November 2020) under different fruit load treatments. Within columns, means with the different letter indicate significant differences ($p < 0.05$) using Duncan's test; bars on the columns represent the standard error of the mean ($n = 12$).

*3.7. Correlation Analysis*

Furthermore, the results of the correlation analysis suggested a significant negative correlation between leaf N levels and flesh sucrose and glucose contents (Table 3), as well as between branch P level and flesh CA content (Table 4); therefore, it could be inferred that under the fruit load treatments set in the present study, a relatively lower leaf N and higher branch P level could effectively promote sucrose and glucose accumulation as well as citric acid degradation in the flesh, thus not only further increasing the TSS/TA ratio, but also changing the proportion of soluble sugar and organic acid components, optimizing the flavor of fruits. However, NPK content in the branches and leaves showed no obvious association with segment hardness and firmness of citrus flesh, in other words, fruit mastication (Table 5).

**Table 3.** Pearson correlation coefficient matrix of NPK content in branches and leaves and fruit soluble sugar content under different fruit load treatments.

|  | Branch-N | Branch-P | Branch-K | Leaves-N | Leaves-P | Leaves-K | Sucrose | Fructose | Glucose |
|---|---|---|---|---|---|---|---|---|---|
| branch-N | 1 | | | | | | | | |
| branch-P | −0.064 | 1 | | | | | | | |
| branch-K | 0.116 | 0.279 | 1 | | | | | | |
| leaves-N | −0.132 | 0.171 | −0.036 | 1 | | | | | |
| leaves-P | 0.326 | −0.204 | −0.124 | −0.317 | 1 | | | | |
| leaves-K | −0.308 | 0.121 | −0.182 | 0.217 | −0.159 | 1 | | | |
| Sucrose | 0.189 | −0.232 | 0.256 | −0.447 * | −0.055 | −0.112 | 1 | | |
| Fructose | 0.129 | −0.306 | 0.187 | −0.418 | −0.093 | −0.034 | 0.965 ** | 1 | |
| Glucose | 0.141 | −0.262 | 0.199 | −0.449 * | −0.062 | −0.032 | 0.972 ** | 0.992 ** | 1 |

\* Indicate a significant correlation at $p < 0.05$ and ** indicate an extremely significant correlation at $p < 0.01$.

**Table 4.** Pearson correlation coefficient matrix of NPK content in branches and leaves and fruit organic acid content under different fruit load treatments.

|  | Branch-N | Branch-P | Branch-K | Leaves-N | Leaves-P | Leaves-K | CA | MA | AsA |
|---|---|---|---|---|---|---|---|---|---|
| branch-N | 1 | | | | | | | | |
| branch-P | −0.064 | 1 | | | | | | | |
| branch-K | 0.116 | 0.279 | 1 | | | | | | |
| leaves-N | −0.132 | 0.171 | −0.036 | 1 | | | | | |
| leaves-P | 0.326 | −0.204 | −0.124 | −0.317 | 1 | | | | |
| leaves-K | −0.308 | 0.119 | −0.182 | 0.217 | −0.159 | 1 | | | |
| CA | −0.113 | −0.450 * | 0.0317 | −0.27 | −0.062 | −0.085 | 1 | | |
| MA | 0.0447 | −0.217 | 0.047 | −0.353 | 0.088 | −0.014 | 0.666 ** | 1 | |
| AsA | −0.057 | −0.366 | −0.013 | −0.207 | −0.092 | −0.027 | 0.832 ** | 0.820 ** | 1 |

\* Indicate a significant correlation at $p < 0.05$ and ** indicate an extremely significant correlation at $p < 0.01$.

**Table 5.** Pearson correlation coefficient matrix of NPK content in branches and leaves and fruit texture indicators under different fruit load treatments.

| | Branch-N | Branch-P | Branch-K | Leaves-N | Leaves-P | Leaves-K | Peel Hardness | Peel Toughness | Segment Hardness | Firmness | Flesh Hardness | Flesh Toughness |
|---|---|---|---|---|---|---|---|---|---|---|---|---|
| branch-N | 1 | | | | | | | | | | | |
| branch-P | −0.06 | 1 | | | | | | | | | | |
| branch-K | 0.12 | 0.28 | 1 | | | | | | | | | |
| leaves-N | −0.13 | 0.17 | −0.04 | 1 | | | | | | | | |
| leaves-P | 0.33 | −0.20 | −0.12 | −0.32 | 1 | | | | | | | |
| leaves-K | −0.31 | 0.12 | −0.18 | 0.22 | −0.16 | 1 | | | | | | |
| Fruit Skin Hardness | −0.11 | 0.03 | 0.09 | 0.16 | 0.16 | 0.07 | 1 | | | | | |
| Fruit Skin Toughness | −0.03 | 0.16 | −0.25 | 0.30 | 0.02 | 0.08 | −0.39 | 1 | | | | |
| Segment Hardness | 0.09 | 0.09 | 0.08 | 0.17 | 0.26 | −0.09 | 0.14 | 0.40 | 1 | | | |
| Firmness | −0.25 | −0.01 | 0.15 | 0.41 | −0.13 | 0.22 | 0.25 | 0.19 | 0.04 | 1 | | |
| Flesh Hardness | 0.28 | 0.06 | −0.30 | 0.16 | 0.18 | 0.08 | −0.25 | 0.64 ** | 0.07 | 0.20 | 1 | |
| Flesh Toughness | 0.25 | 0.40 | −0.15 | −0.11 | 0.10 | 0.12 | −0.16 | 0.06 | 0.01 | −0.20 | 0.40 | 1 |

** indicate an extremely significant correlation at $p < 0.01$.

## 4. Discussion

Fruit thinning was a widely applied practice in various fruits such as peach [20], blueberry [21] and citrus [22], aiming to attain better fruit quality, including larger fruit size and higher TSS content. The present study suggested that citrus fruit TSS increased more under higher levels of tree fruit load (CK) compared to low-load treatments (L-20 and L-30). Furthermore, within the treatment of CK, treatments with relatively lower fruit bearing per branch (CK-1 or CK-2) showed relatively higher TSS content and AWF as compared to that treatment with higher fruit bearing per branch (CK-4) during fruit ripening (Table 2), a similar conclusion was previously drawn by Duarte et al. [22], which suggested that under the relatively higher tree fruit load treatment, less fruit per branch showed higher fruit TSS. In addition, studies on apricots [23] and plums [24] proposed another hypothesis, that TSS content and fruit firmness were not affected by fruit thinning. This might indicate that the regulation mechanisms of fruit development, soluble sugar accumulation and organic acid metabolism by different tree fruit load level showed significant varietal difference.

One of the pathways to further improve fruit quality by delayed harvesting technologies was to extend the stored nutrient supply period of tree bodies and leaves to the fruit, to meet the demand of its continuous TSS further accumulation. Leaves are considered to be the direct source of mineral nutrients for fruit development, and the transportation of stored tree nutrients to leaves and fruit requires sufficient fruit load [21,25], which means that appropriate high tree fruit load might be involved in fruit quality improvement by optimizing the spatial distribution of tree nutrients. It could be concluded from our results (Figure 1) that the N and P nutrients required during fruit ripening periods were mainly supplied by leaves; similar results were reported by Goldschmidt and Koch [26] in citrus, that sucrose is the major sugar translocated from the leaves to the fruit. While the K nutrients came from both leaves and branches, the relatively low tree load was not conducive to promoting the transportation of N and K nutrients from the leaves to the fruit, especially during the later stage of fruit ripening. This might be one of the reasons why the fruit TSS content cannot be further improved under low-load treatment, even if delayed harvesting technologies are implemented.

Both soluble sugar accumulation and organic acid degradation are important processes for sugar/acid quality formation. Previous studies have suggested that puffing may be caused by excessive soluble sugar accumulation in the peel during the later stage of fruit ripening; either excessive transportation and accumulation of soluble sugar from the flesh to the peel [27] or greater respiration consumption of soluble sugar in the peel than that of the peel [28] are possible incentives. However, this present study showed that during the delayed harvest time period, fruits under the low-load treatments showed an even greater increases in flesh fructose and glucose as compared to that under the treatment of CK, while in the opposite, flesh sucrose under the treatment of CK showed a greater increase than that of low-load treatments, which suggested that the fruit TSS further increase under delayed harvest was mainly due to the continuous increase of flesh sucrose content. Meanwhile, peel sucrose content under treatment of CK and L-20 kept stable, while low-load treatment of L-30 resulted in significantly decrease. Therefore, this present study puts forward a different possibility that soluble sugar accumulation in the peel and the flesh were relatively independent (Figure 2),

Although citric acid is the major organic acid of citrus fruit, accounting for 90% of the total acids, malic acid also accumulates to some extent during the maturation period of citruses [3,9,29]. This present study suggested that lower load treatment resulted in a greater decrease in flesh MA content, but had no significant effect on flesh CA content (Figure 3). Zhang et al. [30] reported that relatively low tree fruit load might increase the utilization of K nutrients, thereby promoting the degradation of MA during apple fruit development. Zhou et al. [19] put forward a different point of view, that the development of the TSS/TA ratio of 'Flame' citrus during fruit ripening was mainly achieved by adjusting the degradation rate of CA, but flesh sucrose or MA content were not significantly

affected. These varied results indicated that different levels of tree fruit load may change the metabolism rules of organic acid degradation in different tree species and cultivars.

Previous studies have provided several different views on the mechanisms responsible for inducing puffiness; for example, Pham et al. [15] reported that sweet orange puffiness resulted from albedo breakdown associated with Ca deficiency. Moreover, secretory cavity rupture due to excessive precipitation [31] or too much nitrogen fertilizer application [10] might also be an important cause of puffiness. In addition, vesicle collapse induced by oxidative stress was also believed to be associated with puffiness [9,12]. However, this present study suggested an alternative viewpoint, that puffiness might be primarily induced by the lignification of parenchyma cells around secretory cavities, which may affect the structural toughness of the peel flavedo cell wall (Figure 4). AsA is one of the nonenzymatic antioxidant protectants of the cell membrane structure, which can prevent the lignification of parenchyma cells [5]. There are other kinds of antioxidants in citrus peels, such as phenolics [11], flavonoids and carotenoids [12], which also have a similar effect to AsA in reducing puffiness.

The organoleptic quality of citrus fruits is easily overlooked; it is mainly determined by sensory texture perceived in citrus segment membrane properties and firmness. The fruit mastication is one of the most important factors to evaluate citrus organoleptic quality, and is related mainly with dietary fiber, especially pectin [32,33]. Segment hardness was closely related with composition and content of the cell wall pectin, and the firmness is mainly affected by the arrangement compactness, water content and membrane hardness of juice sacs [34]. It is generally believed that both segment hardness and firmness have a negative correlation with fruit mastication. In this present study, segment hardness and firmness showed no obvious difference within treatments of CK, but these were relatively higher and varied significantly in L-20 and L-30, which suggested that the effects of tree fruit load on fruit mastication showed a more significant priority than fruit bearing per branch. Besides, results of correlation analysis performed showed that tree NPK nutrient levels demonstrated no significant association with fruit texture (Table 5). Similar results were reported in *C. ladanifer* [25] and in blueberries [21], that the flesh texture was more obviously affected by tree fruit load and fruit-bearing per branch than the tree nutrient levels.

Meanwhile, results of correlation analysis also tell that relatively lower leaf N and higher branch P can promote accumulation of sucrose and glucose, as well as degradation of citric acid, respectively (Tables 3 and 4). Similar results were reported by Ilan et al. [28], that in pummelo, excessive N application not only slows down the accumulation of soluble sugar and delays the ripening of the fruit, but also causes flesh hardening. Zhang et al. [30] found that K fertilizer could promote fructose, sorbitol, glucose, and sucrose accumulation by affecting the activities of sugar synthesis-related enzymes. However, in the present study, the correlation between K nutrient and soluble sugar and organic acid metabolism was not significant (Tables 3 and 4). These results indicated that the varied effects of tree nutrition regulation on the soluble sugar and organic acid metabolism and flavor quality formation may be caused by differences in tree varieties and experimental treatments. Moreover, Zheng et al. [34] reported that in 'Nanfeng' tangerines, fruit segment shear force showed a significantly negative correlation with flesh N and P contents. Therefore, the regulatory mechanism whereby the tree fruit load affects the NPK contents of fruits and its influence on fruit mastication needs to be further explored.

## 5. Conclusions

Taken together, our results demonstrated that relatively higher tree fruit load and higher fruit bearing per branch could promote the flavor quality of fruits through enhancing the soluble sugar accumulation as well as optimizing the ratio of different organic acid components. Although higher tree fruit load could significantly improve the utilization of tree stored K nutrients, the fruit sucrose synthesis and citric acid degradation process are more closely related with leaf N and branch P nutrient levels, respectively. Low tree fruit load is not only disadvantageous to flavor improvement (especially sucrose accumulation

of citrus fruits), but also might cause serious puffiness. There are two possible primary mechanisms of puffiness: the first is that the toughness and tensile strength of the peel cell wall were obviously weakened due to the lignification of parenchyma cells around the secretory cavities of the peel; the other is premature senescence of the peel caused by AsA deficiency-induced oxidative damage. These two phenomena are especially obvious under a relatively low tree fruit load.

**Author Contributions:** Data curation, J.Z.; Funding acquisition, F.J.; Project administration, F.J.; Resources, H.A.; Supervision, Z.X.; Writing—original draft, F.X.; Writing—review & editing, Z.X.; All authors have read and agreed to the published version of the manuscript.

**Funding:** This research was funded by Shanghai Science and Technology & Agriculture Promoting Project for Funding (Study on the key techniques of organic cultivation of Citrus), grant number 2019-02-08-00-16-F01126.

**Institutional Review Board Statement:** The study did not involve humans or animals.

**Informed Consent Statement:** The study did not involve humans.

**Data Availability Statement:** The data required to reproduce these finding cannot be shared at this time as the data also forms part of an ongoing study.

**Conflicts of Interest:** The authors declare no conflict of interest.

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
