# Peer review of "Effects of Fruit Load on Sugar/Acid Quality and Puffiness of Delayed-Harvest Citrus"

_horticulturae, doi:10.3390/horticulturae7070189_

Round 1

Reviewer 1 Report

Your manuscript needs this attached reference incorporated into your discussion of results: https://www.frontiersin.org/articles/10.3389/fpls.2019.01167/full

I cannot make the PDF attach so I am giving the reference.

Author Response

Answers to Reviewer 1 ’s comment

Comments and Suggestions for Authors

Your manuscript needs this attached reference incorporated into your discussion of results: https://www.frontiersin.org/articles/10.3389/fpls.2019.01167/full

I cannot make the PDF attach so I am giving the reference.

Submission Date

26 May 2021

Date of this review

05 Jun 2021 18:30:04

Answer: we added the Review <Sadka et al. (2019) Primary Metabolism in Citrus Fruit as Affected by Its Unique Structure> into the reference list and carefully revised the discussion part of this manuscript. Thank you for your valuable comments.

Reviewer 2 Report

The topic of the article is important and the results must be published.

The following are some specific notes on the text, proposing amendments.

Line 21: The authors never explain the meaning of "CK"

Line 41: The reference [3] is completely wrong. This article refer to Tadeo et al 2008 and presents another division into phases not these period and not these descriptions.

Line 63: Currently it seems to be more correct to use "Citrus trifoliata".

Line 164-165: The graphics are so small and blurry that I can't even evaluate them.

Line 191-192: The graphics are so small and blurry that I can't even evaluate them.

Line 198: The abbreviations "CA" and "MA" are not obvious. They were explained before, but, as they are not usual and obvious, they make it difficult to read the text.

Line 217-218: The graphics are so small and blurry that I can't even evaluate them.

Line 229-231: The photos are of poor quality and the bars are not visible.

Line 240 (Fig 5): Too small graphs.

Lines 264- 272: TSS strongly depends on the size of the fruit (https://www.researchgate.net/publication/269168509_Thinning_of_'Esbal'_clementine_with_24-dichlorophenoxyacetic_acid_Influence_on_yield_fruit_size_and_fruit_quality); It would be important to know the size of the analyzed fruits. On the other hand, there are other factors, such as thinning with auxins, which can influence the TSS in fruits of approximately equal size (Journal of the American Society for Horticultural Science, 131 (5): 586-592.).

There are more suggestions in the attached file.

Author Response

Answers to Reviewer 2’s comments:

Line 21: The authors never explain the meaning of "CK"

Answer: The present study set three tree fruit load level: the control (CK, 120 fruits per tree), 20 fruits per tree (L-20), and 30 fruits per tree (L-30). CK (Line 68-69). In order not to exceed the word limit of Abstract, we changed “the CK” (Line 23) to “relative higher load” and the explaination of the meaning of “CK” was performed in “Material and methods”.

Line 41: The reference [3] is completely wrong. This article refer to Tadeo et al 2008 and presents another division into phases not these period and not these descriptions.

Answer: we studied the paper<Tadeo et al. (2008) Molecular Physiology of Development and Quality of Citrus> thoroughly, and we confirmed that the method of dividing citrus development period in our manuscript is completely different from that that described in the reference. In fact, the present study was mainly focused on the later stage of fruit development, especially the period of flavor quality formation (rapid accumulation of soluble sugar and degradation of organic acid). Actually, we tried to divide the fruit rapid growing and fruit maturing period into three parts. so in the revised version of our manuscript, so we corrected this mistake and rewrote this part according to the comments of the reviewers and the experimental design. 

Line 63: Currently it seems to be more correct to use "Citrus trifoliata".

Answer: we corrected this words according to this comments.

Line 164-165: The graphics are so small and blurry that I can't even evaluate them.

Line 191-192: The graphics are so small and blurry that I can't even evaluate them.

Line 217-218: The graphics are so small and blurry that I can't even evaluate them.

Line 229-231: The photos are of poor quality and the bars are not visible.

Line 240 (Fig 5): Too small graphs.

Answer: we redrawn and re-upload the original format of these Figures as an attachment with the manuscirpit (revised version) according to the comments of the reviewers.

Line 198: The abbreviations "CA" and "MA" are not obvious. They were explained before, but, as they are not usual and obvious, they make it difficult to read the text.

Answer: we corrected these two abbreviations “CA” and “MA” into “citric acid” and “malic acid” according to the comments of the reviewers.

Lines 264- 272: TSS strongly depends on the size of the fruit (https://www.researchgate.net/publication/269168509_Thinning_of_'Esbal'_clementine_with_24-dichlorophenoxyacetic_acid_Influence_on_yield_fruit_size_and_fruit_quality); It would be important to know the size of the analyzed fruits. On the other hand, there are other factors, such as thinning with auxins, which can influence the TSS in fruits of approximately equal size (Journal of the American Society for Horticultural Science, 131 (5): 586-592.).

There are more suggestions in the attached file.

Answer: we added this paper into the reference list and rewrite the first paragraph of Discussion according to the comments of the reviewers.

What is more, the grammar mistake and spelling mistake were all corrected according to the suggestions in the attached file.

Reviewer 3 Report

Major concerns:

The vocabulary “Flavor” might be inappropriate in the title. In addition to the sugar/acid, contents such as some secondary metabolites influence the quality of a fruit flavor.

The resolution of Figure 1-4 is quite low; In table 2, no letters mark the difference.  A manuscript with higher resolution figures is needed.

Minor concerns:

Line 189, a typo, double “that”

What does the phrase “fruit mastication” mean?

Line 353, no verb in the sentence “Low tree fruit load not only disadvantageous to flavor improvement”

Author Response

Answers to Reviewer 3 ’s comments:

Major concerns:

The vocabulary “Flavor” might be inappropriate in the title. In addition to the sugar/acid, contents such as some secondary metabolites influence the quality of a fruit flavor.

Answer: the present study mainly focused on the metabolism of soluble sugars and organic acids and changes of TSS/TA ratio during the later development period of citrus fruits (fruit ripening period) , according to the word limit of title, How about the new title “Effects of fruit load on quality and puffiness of delayed-harvest citrus”? or “Effects of fruit load on sweetness and puffiness of delayed-harvest citrus”?

The resolution of Figure 1-4 is quite low; In table 2, no letters mark the difference.  A manuscript with higher resolution figures is needed.

Answer: we re-drawn the Figure 1-4 and marked the difference in Table 2 according to the comments of reviewers

Minor concerns:

Line 189, a typo, double “that”

Answer: we deleted one of “that” according to the comments of reviewers.

What does the phrase “fruit mastication” mean?

Answer: The fruit mastication character is an important factor to evaluate fruit quality of citrus, and is related mainly with dietary fiber, especially pectin. Cell wall is composed of cellulose, hemicellulose, pectin, and sometimes lignin. Pectin is one of the important compounds that harden the cell wall. When it is degraded by enzymes during fruit maturation, tissue softens gradually. Cellulose and hemicellulose are also degraded by cellulase during this period.The fruit pulp mastication characteristics can be quantified using a fruit texture analyze (Result 3.6).

Line 353, no verb in the sentence “Low tree fruit load not only disadvantageous to flavor improvement”

Answer: we have corrected this mistake and rewrite this sentence according to the comments of reviewers. 

Thank you for your valuable comments!

Round 2

Reviewer 3 Report

Xu et al. explored the correlation between the NPK load and citrus quality (sugar/acid content and puffiness) in the manuscript. The results could provide information in controlling the citrus fruit quality, especially in the delayed harvest. In general, the manuscript is easy to follow. Still, I found a few concerns that might influence the quality of the manuscript for publication in the current version.

Major concerns,
1, The assay lasted two years (2019-2020). For a biological assay, the period is relatively long for sample collection. However, it seems some data from 2019 and 2020 was not consistent (as stated in lines 166-174). It is better to add another year-round data to solidify the conclusion, or at least the authors should give simple explanations in the results/discussion regarding the NP content changes.
2, The fertilizer was applied every two weeks. In Table 1, please add more details about the way (time) of soil collection. "Before/After harvest" is not specific and would confuse.
3, Please mark the deformed/ruptured cavities in figure 4.

Minor concerns,

For the title, the word “quality” is even larger than “Flavor”. I suggest specify the word as “sugar/acid quality”.

Line 13, “reducing” should change to “reduce”

Line 14, “as well as to”, delete “to”.

Line 15, “an experiment was” to “experiments were”

Line 24, “ASA” should use a full name in the abstract.

Line 24, adding, might be “a or the” primary cause.

Line 213 214 217, “sucrose, glucose and fructose” to “sucrose, glucose, and fructose”

Line 359 360, citric acid and malic acid are not the first time used in the manuscript. The abbreviations should move forward.

Author Response

Reviewer’s comments:

Major concerns

1, The assay lasted two years (2019-2020). For a biological assay, the period is relatively long for sample collection. However, it seems some data from 2019 and 2020 was not consistent (as stated in lines 166-174). It is better to add another year-round data to solidify the conclusion, or at least the authors should give simple explanations in the results/discussion regarding the NP content changes.

Answer: we added some new explanation to discuss the NP content changes in both the part of Results and Discussion, and rewrite the relevant part of the manuscript (Results 3.2).

In summary, leaf N showed a similar decreasing trend in 2019 and 2020, and the decrements under relatively higher tree fruit load/higher fruit bearing per single branch were greater; the reasons might be as follows: (1) leaf was one of the direct N source for fruit development; (2) higher tree fruit load/higher fruit bearing per single branch formed greater sink strength to attract more N nutrients from leaf to fruits. (3) branch N was a supplement for leaf N, so the change of branch N content was relatively laggard as compared to that of leaf, if nitrogen was sufficient for the fruit development, the fruits will still get N nutrients from the leaf preferentially, Therefore, we can conclude that this higher tree fruit load level for two consecutive years caused too much consumption of tree stored nutrients, especially branch NP, so we speculate that the inconsistent data in the two consecutive years may be due to the fertilization and the overall change of tree nutrient level.

2, The fertilizer was applied every two weeks. In Table 1, please add more details about the way (time) of soil collection. "Before/After harvest" is not specific and would confuse.

Answer: we added more information about the time point of soil collection: Before harvest (mid September): during the period of rapid accumulation of soluble sugar/organic acid of fruits; After havest (late-November): during the time period of that after fruit harvest and before the fertillization in winter.(L166-168)

3, Please mark the deformed/ruptured cavities in figure 4.

Answer: we marked the normal secretory cavities with blue arrows and the deformed/ruptured cavities with red arrows, and we updated the Figure 4 in the revised versionin of our manuscript.

Minor concerns

For the title, the word “quality” is even larger than “Flavor”. I suggest specify the word as “sugar/acid quality”.

Answer: we changed the title to “Effects of fruit load on sugar/acid quality and puffiness of delayed-harvest citrus

Line 13, “reducing” should change to “reduce”

Answer: we corrected “reducing” to “reduce” and many thanks for helping us to correct the gramma mistake in this manuscript.

Line 14, “as well as to”, delete “to”.

Answer: we delete “to” and many thanks for helping us to correct the gramma mistake in this manuscript.

Line 15, “an experiment was” to “experiments were”

Answer: we corrected “an experiment was” to “experiments were”

Line 24, “ASA” should use a full name in the abstract.

Answer: we use “ascorbic acid” instead of the “AsA” in the abstract.

Line 24, adding, might be “a or the” primary cause.

Answer: we add an “the” before “primary cause for puffiness under low-load treatments”

Line 213 214 217, “sucrose, glucose and fructose” to “sucrose, glucose, and fructose”

Answer: we add “ , ” between “sucrose, glucose” and “and fructose” (L237 in revised version)

Line 359 360, citric acid and malic acid are not the first time used in the manuscript. The abbreviations should move forward.

Answer: we deleted the abbreviations here and moved them forward to L 132(In the part of Material and Methods).
